# Performance Degradation of Large-Sized Asphalt Mixture Specimen under Heavy Load and Its Affecting Factors Using Multifunctional Pavement Material Tester

**DOI:** 10.3390/ma12233814

**Published:** 2019-11-20

**Authors:** Jiasheng Li, Jianying Yu, Jun Xie, Qunshan Ye

**Affiliations:** 1State Key Laboratory of Silicate Materials of Architectures, Wuhan University of Technology, Wuhan 430070, China; lijiasheng@whut.edu.cn (J.L.); jyyu@whut.edu.cn (J.Y.); 2Key Laboratory of Road Structure and Material of Ministry of Transport (Changsha), Changsha University of Science & Technology, Changsha 410114, China; yeqs@whut.edu.cn

**Keywords:** asphalt mixture, heavy load, permanent deformation, skid resistance, aggregate sieving

## Abstract

With the increase of heavy traffic transportation, it is meaningful to study the performance change of pavements under heavy loads. To study the development of asphalt mixture under heavy load, an AC-13 asphalt mixture was prepared and a large-sized specimen wheel tracking test was conducted. Samples from different periods were extracted to research the influence of the cumulative load times under heavy load on asphalt mixtures. It was found that there were different variation rules on the performance of the asphalt mixture under different load conditions. The failure time on skid resistance was predicted by the fitting curve. Heavy load conditions would greatly speed up the failure of skid resistance. According to the sieving results, the stable gradation of 10–16 mm, 5–10 mm, 3–5 mm, and 0–3 mm after heavy load grinding was 10%, 37%, 23%, and 30%. It was found that the content of 10–16 mm had a significant correlation with the permanent deformation of the asphalt mixture, while there were no significant correlation between aggregate and skid resistance.

## 1. Introduction

With the development of road transport, the highway sustains enormous traffic volume, especially with many overloaded trucks. The wheel rolling action of overloaded trucks’ is one of the major causes of permanent deformation and a reduction of pavement performance of asphalt mixture surface layer [1,2]. Heavy loads increase the contact and resistance between the tire and the ground, which accelerate the polishing effect of pavement surface, resulting in the decline of the skid resistance of the asphalt mixture and increasing the risk for traffic safety [3,4].

Driven by economic interests and other factors, the heavy trucks on the highway are overloaded frequently in China. At present, the average axle load of heavy vehicles is about 12–18 t in China, and the tire pressure is about 0.8–1.1 MPa. With different tire pressure, tread pattern, and load, the effective grounding area of the tire is about 25,000–70,000 mm^2^. The replacement diameter of wheels is about 175–300 mm. The actual axle load of trucks takes 50%–100% more than rated in general and the pressure could be up to 1.7 MPa, which has brought serious impact on the road pavement [5]. Normally, the newly built pavement can easily meet the requirements of strength and skid resistance. However, with the compacting, wearing, and polishing of the vehicle, many pavements lose their stable structure, followed by a rapid decrease in their strength and anti-skid ability [6].

Permanent deformation directly reflects the riding comfort and safety and service life of an asphalt pavement [4]. Texture depth (TD) and British pendulum number (BPN) can reflect the surface texture and skid resistance of the pavement indirectly [7,8]. Normally, asphalt pavements with higher TD and lower BPN have superior skid resistance [9,10]. The test of TD and BPN can compendiously describe the macro texture depth and friction coefficient, respectively [11]. These brief measurement methods will provide significant information for policymakers to optimize pavements [12,13].

The majority of studies on the performance of the asphalt mixture has focused on the selection of aggregates and the gradation of mixture, while there has been relatively few research on the influence of heavy load [7,14,15]. Considering the characteristic of heavy axle load in modern transportation, the objective of this study was to measure the mechanics and skid resistance after various heavy loads on asphalt concrete (AC) pavements. The mechanical performance was measured by permanent deformation and the skid resistance was measured by TD and BPN through the sand patch method and British pendulum. Furthermore, the aggregate gradation after heavy load was analyzed.

## 2. Materials and Experimental Program

Crushed basalt was used as the coarse and fine aggregate to prepare AC-13 asphalt mixtures. Basalt is often selected as the aggregate of an asphalt mixture in upper pavements because of its high strength, durability, and good adhesion to bitumen. Aggregates passing percentages at sieve of sizes 16, 9.5, 4.75, and 2.36 mm were 100%, 72%, 44%, and 30%, respectively. The average density of the used basalt aggregates was 2.983 g/cm^3^, Los Angeles abrasion was 20.4%, the polished stone value was 64, and the crushed stone value was 19.5%. The gradation curve is shown in Figure 1.

The physical properties of Styrene-butadiene-styrene block copolymer (SBS) asphalt binder were illustrated in Table 1. The asphalt mixture was prepared according to specification JTG E42-2005 [16]. The percentage of AC-13 asphalt mixture components were 4.4% SBS bitumen, 26.8% 10–16 mm basalt, 26.8% 5–10 mm, 13.4% 3–5 mm, 24.8% 0–3 mm, and 3.8% basalt powder filler. The air voids of the asphalt mixture was measured to be 4.5% when designing the mix proportion. The AC-13 asphalt mixtures were used for the upper layer of the pavement.

The various loading tests were conducted in a multifunctional pavement material tester (MPT) developed by Wuhan University of Technology (Figure 2). Its functional part was a rubber wheel with a width of 40 mm that moved on two steel rails. The contact area was 1200 mm^2^. The load and speed of the flat rubber wheel was controlled by MPT to apply load on specimens, ranging from 0 to 30 MPa. The temperature inside was controlled by heating tubes and a water cooling circulation. Furthermore, there were additional modules that controlled UV, oxygen concentration, humidity, etc.

Aggregates and SBS bitumen were preheated to 165 °C and compacted by a steel roller. To fully simulate the pavement service environment [17,18], large-sized specimens with a length of 1000 mm, a width of 500 mm, and a height of 40 mm were used in MPT (Figure 3). The finished specimens were left in room temperature for 24 h and conditioned in MPT for at least 6 h before test. The tests were carried out at 60 °C and 40% humidity, and the wheel speed was 12 cycle/min [16,19,20].

For each specimen, 10 aged core samples were extracted after 12 h, 36 h, 60 h, 84 h, and 108 h. The tested specimen was cooled to 15 °C in MPT. Then, the core samples were extracted by a core drilling machine. Two samples were taken at each time point for error reduction. The core sample was a cylinder with a diameter of 100 mm and a height of 40 mm. The specimens were refilled to make up the holes with the same asphalt mixture after drilling so that the next tests could proceed smoothly. Next, the specimen was heated to 60 °C for 6 h in MPT for subsequent tests. The permanent deformation of each sample was measured by a displacement sensor. Then, the unaffected part of the core sample was cut off to prepare the samples for other tests.

The measurement of TD and BPN of samples was conducted with the sand patch method and British pendulum according to specification JTG E42-2005 [16]. Before testing the BPN, samples were conditioned in 20 °C and moistened by water. After the BPN and TD tests, samples were dissolved by toluene to separate aggregates from the asphalt binder. Then, the aggregates was sieved to observe the variation of aggregates after heavy load.

The heavy load testing program for the upper layer of pavement of the AC-13 basalt asphalt mixture is shown in Table 2. The load was determined from a survey report of Chinese highway transportation to simulate the load that the overloaded truck brought to the asphalt pavement. A pressure of 0.7 MPa was selected to represent normal wheel load and 0.9 MPa, 1.1 MPa, and 1.4 MPa were selected to simulate the overload environment [1]. Each load corresponded to a series number, and each sample’s label indicated its load and duration.

The time–temperature superposition and thermo-rheological simplicity principles belong to the basic assumptions in the linear viscoelastic analysis [21,22]. Therefore, it is commonly used in the analysis of asphalt mixtures [23]. The WLF (Williams–Landel–Ferry) equation (Equation (1)) is defined using substantial experimental data based on the free volume theory by Williams, Landel, and Ferry [24].
(1)$${\mathrm{lg}\alpha }_{T}=\frac{{-C}_{1}{(T-T}_{0})}{C_{2}{+(T-T}_{0})}$$
where C_1_ and C_2_ are constants of the corresponding materials; T is the shift temperature; T_0_ is the reference temperature; ${\mathrm{lg}\alpha }_{T}{\;=\;\mathrm{lgt}}_{0}\;–\;\mathrm{lgt}$; and t_0_ and t are the corresponding times of T_0_ and T.

Equation (1) can be used to determine the corresponding time in real conditions by plugging in the constants C_1_ and C_2_ of the asphalt mixture [25] and the average temperature in central China. For the asphalt mixture, C_1_ and C_2_ were 38.46 and 316.35, respectively. T_0_ took the average annual road surface temperature of 327.75 K (54.6 °C) in central China (Wuhan). T took the test temperature of 333.15 K (60 °C) in MPT. It was calculated that one day in the MPT equaled 4.65 days in real conditions. It was assumed that the width of the tire was 215 mm, the flat ratio was 70%, and speed was 90 km/h. It could be calculated that the average contact time between the tires and ground was 0.006 s. According to JTG D50-2006, the traffic volume of a heavy load road was more than 1500 times per day with medium and large trucks and buses [26]. The vehicle model adopted a double rectangular uniformly distributed load. The coefficient of lanes was 0.45. It could be calculated that the contact time per day was 16.1 s. According to the parameter settings in MPT, the contact time between the tire and specimen in MPT could convert to 502.6 s/h in real conditions. Therefore, 1 h in MPT could simulate 31 days in real conditions. This experiment could simulate the pavement condition of up to nine years in real conditions.

## 3. Experimental Results and Discussion

### 3.1. Permanent Deformation Results

Figure 4 shows the effects of various heavy loads on the development of rutting of the AC-13 pavement asphalt mixture. Four series were selected to represent different load levels and each series was constituted by five periods of cumulative cycles.

Normally, the development of permanent deformation of an asphalt mixture is divided into three stages: (1) Initial stage: the permanent deformation increases rapidly, but the growth rate decreases gradually; (2) Steady stage: the permanent deformation increases steadily, and the growth rate remains the same; and (3) Destroy stage: the permanent deformation and the growth increase rapidly until failure [27].

As shown in Figure 4, C and D revealed two distinct rutting stages while A and B rose steadily. The permanent deformation of C and D could be divided into two stages: the initial stage (before 43,000 cycles) and the steady stage (43,000–80,000 cycles). However, the destroy stage was not observed. This indicated that the asphalt mixture would not go into the destroy stage in the service of nine years. The permanent deformation rose sharply in the initial stage and then became almost constant in the steady stage. The analysis of experimental data showed that 90% permanent deformation had been accomplished in the initial stage for lines C and D. For lines A and B, the increase of permanent deformation was gentle and without obvious stage. The final value of A and B was 75% of D, equal to D with 26,000 cycles.

According to JTJ073.2-2001 [28], the allowable value of deformation is 15 mm in highways. Once the deformation exceeded 15 mm, the pavement took the risk of failure and needed major repair. This indicates that pavements under normal load or slight overload will not run over the allowable value in the service of nine years. However, the pavement under heavy load would exceed the allowable deformation within 45,000 cycles.

The wheel tracking slope (WTS) can be used to describe the increase in permanent deformation [29]: WTS = (h_cycle1_−h_cycle2_)/(cycle1−cycle2), where h_cycle_ is the depth of deformation in the cycle. According to the data, after 60,000 cycles of the tests, the WTS of A, B, C, D were calculated as 0.89 mm/(10^4^ cycles), 0.92 mm/(10^4^ cycles), 0.15 mm/(10^4^ cycles), and 0.40 mm/(10^4^ cycles), respectively. The WTS of A and B were much higher than C and D. This illustrates that the growth of deformation had slowed down in C and D, but remained stable in A and B. Therefore, the structure of C and D went into a stable state, while the structure of A and B were still changing.

The above results indicate that there was a steady growth in the permanent deformation of the asphalt pavement under some critical pressure value. According to the results of this experiment, this critical load value was nearly 1.0 MPa, depending on the performance of the aggregates used. Once the vehicle load was beyond the critical load value, the permanent deformation of the asphalt pavement was going to develop rapidly to reach the majority of the final value. Then, the following development of permanent deformation could be very slow-growing. The destruction of stable structure under high load in the initial stage led to the sharp increase in deformation. As shown in Figure 5, the aggregates were crushed into pieces and reformed a new stable structure. Once the new stable structure took shape, the rapid increase in permanent deformation would slow down.

### 3.2. Skid Resistance Results

Figure 6 and Figure 7 show the variation in the skid resistance of the asphalt pavement under heavy load. The initial BPN and TD of samples were 75.5 and 1.873 mm, respectively. 

As shown in Figure 6, the BPN of the asphalt mixture under various load decreased with the increase in cumulative cycles. At the beginning of the test, BPN decreased rapidly and stabilized gradually with the increase of cumulative cycles. The final BPN dropped by 23%–31% to a varying degree. With the increase in wheel load, the decline in the mixture became faster and the abrasion of aggregates became more severe. The mixture under 1.4 MPa had the maximum degradation range and the minimum final value.

It was found that after 4000~5000 times of wheel grinding, the asphalt mixture on the surface was compacted to expose the edges of the inner aggregates because of densification. After that, the aggregate corners and edges without mixture protection were gradually destroyed by the wheel. The skid resistance began to decline rapidly after this point. The mixture under 1.4 MPa was worn severely under the wheel load. There were obviously crushed aggregates and bleeding on the surface of the mixture under 1.1 MPa and 1.4 MPa. There were no obviously crushed aggregates on the surface of the mixture under 0.7 MPa and 0.9 MPa, but bleeding was observed to different degrees.

According to JTJ073.2-2001 [28], the allowable BPN of highway should be higher than 45. The fitting curve for BPN to the cumulative cycles under 0.7 MPa was plotted with the model of power function by software Statistical Product and Service Solutions (SPSS). According to the fitting curve, the approximate cumulative cycles under 0.7 MPa required to degrade the BPN to 45 was estimated to be 4.09 million. The cumulative cycles under 0.9 MPa, 1.1 MPa, and 1.4 MPa could be estimated as 1.26 million, 0.55 million, and 0.34 million, respectively with the similar approach. The simulated results showed that skid resistance could be maintained for a long time under normal load. However, with the increase in wheel load, the skid resistance disappeared quickly.

As shown in Figure 7, the change rule of TD was similar to the BPN. In the initial stage of heavy wheel load, the asphalt mixture was compacted to expose the corners and edges of aggregates on the surface. The destruction of the surface structure led to the quick decline of TD. Then the corners of aggregates were ground off and the aggregates was crushed into pieces by heavy load gradually, leading to further decline of TD in the later stage. Eventually, the decrease of TD tended to balance with the polishing of the mixture surface. Overall, TD decreased by 40–60% with various wheel loads, where the higher load made the faster decline and the lower final value of TD.

According to JTG D50-2006 [26], the allowable TD for an asphalt mixture should be higher than 0.55 mm. The same analytical method as BPN was adopted to estimate the maximum cumulative cycles of 0.7 MPa, 0.9 MPa, 1.1 MPa, and 1.4 MPa were 0.88 million, 0.39 million, 0.23 million, and 0.12 million, respectively. This simulation illustrates that the TD of the asphalt mixture failed faster than BPN. The maximum cumulative cycles under heavy load would be reduced by more than 80%, and a heavy load condition would greatly speed up the failure in skid resistance.

### 3.3. Sieving Results

The sieving results of the ground mixture are shown in Table 3. Due to the loss of fine aggregates and mineral powder filler during dissolution and filtration, the percentage of 0–3 mm parts was compensated by 3%.

As shown in Table 3, the 0–3 mm and filler part of the aggregates changed insignificantly in the grinding process. Considering the loss in separation, it may be that the 0–3 mm and filler part had no loss or increase in the whole process. The content of 10–16 mm decreased continuously, while the content of 5–10 mm and 3–5 mm increased with the progress of the tests. Figure 8 showed the change of the other parts in the grinding process. The Y-axis in Figure 8 represents the percentage of the aggregate in the ground asphalt mixture. Figure 8A shows the 10–16 mm loss, and B and C show the rise in 5–10 mm and 3–5 mm, respectively.

As shown in Figure 8A, the 10–16 mm part was greatly affected by the load and cumulative cycles. In the whole process, the loss of 10–16 mm was more than 15% of the total. The decline of the 10–16 mm was slow under lower loads (0.7 MPa and 0.9 MPa), while the decline under a higher load (1.1 MPa and 1.4 MPa) was faster. Relatively, the changes in the 5–10 mm and 3–5 mm parts were less than 10–16 mm, owing to both additions crushed from the upper aggregates and loss crushed into smaller pieces. Furthermore, the rise in 5–10 mm roughly equaled the rise in 3–5 mm. Overall, the loss of 10–16 mm approximately equaled the rise of the 5–10 mm and 3–5 mm. Ignoring the repeated increase or decrease, it could be considered that the effect of heavy load on the AC-13 asphalt mixture was to divide the 10–16 mm part half into 5–10 mm, and the other half into 3–5 mm.

The curve under the higher load presented two stages, the same as the curve of permanent deformation under high load. It illustrated that the deformation of the pavement under high load was related to the change in the aggregate gradation. According to Table 3, mixtures under 1.1 MPa and 1.4 MPa had similar final aggregate gradations after a long period of grinding. The new aggregate gradation of 10–16 mm, 5–10 mm, 3–5 mm, and 0–3 mm was approximately 10%, 37%, 23%, and 30%, respectively. This suggests that C and D formed a similar stable skeleton structure, while the skeleton structures of A and B were still changing. This result was consistent with the conclusion of permanent deformation. This demonstrates that the aggregates would be crushed into smaller pieces and reform a stable structure after high load grinding.

The gradation curve after heavy load grinding is shown in Figure 9. Although the hybrid mixture curve was close to the upper limit, it was still between the upper and lower limit.

### 3.4. Correlation Analysis

To investigate the cause of the performance change of the asphalt mixture, correlation analysis between the content of different aggregates and mixture performance was conducted by SPSS. The Pearson correlation coefficient (PCC) and the significance level (Sig.) are computed and listed in Table 4. The PCC is a measure of the linear correlation between two variables, X and Y. The PCC has a value between 1 and -1, where 1 is total positive correlation, 0 is no correlation, and -1 is total negative correlation. The PCC can be computed by Equation (2) [30].
(2)$$r_{\mathrm{xy}}=\frac{\sum_{i=1}^{n}{(x}_{i}{\;-\;\bar{x})(y}_{i}-\;\bar{y})}{\sqrt{\sum_{i=1}^{n}{\left(\mathrm{xi}-\bar{x}\right)}^{2}}\sqrt{\sum_{i=1}^{n}{\left(\mathrm{yi}-\bar{y}\right)}^{2}}}$$
where n is sample size; x_i_, y_i_ are the individual sample; $\bar{x}=\;\frac{1}{n}\overset{n}{\underset{i=1}{\sum }}x_{i}$, analogously for $\bar{y}$.

As shown in Table 4, for aggregates in the three sizes of 10–16 mm, 5–10 mm, and 3–5 mm, the correlation between 10–16 mm and mixture performance was the highest, followed by 3–5 mm and 5–10 mm. Permanent deformation had the highest correlation with aggregates among the three performance indexes. All the aggregates could pass the 0.05 significance test on permanent deformation. That indicated there were relationships between aggregate gradation and permanent deformation. Furthermore, the 10–16 mm aggregate could pass the 0.01 significance test on deformation. This indicates that the 10–16 mm aggregate had the most significant correlation with permanent deformation. This implies that the content of 10–16 mm in pavements could be used to measure its performance. However, TD had the lowest correlation with aggregates and most of them failed the 0.05 significance test. Relevant data showed that TD and BPN had a good correlation, and the correlation coefficient under various loads was above 0.97. This indicates that there was no significant relationship between the skid resistance of the asphalt mixture and the aggregate gradation, but there was a significant relationship between BPN and TD. This illustrates that the rutting performance was related to the aggregate and its gradation, while the skid resistance was related to the shapes, edges, and corners of the aggregate on the surface.

With the increase in load, the correlation between permanent deformation and aggregates gradually decreased. This indicates that the permanent deformation had a higher correlation with aggregates under low load.

## 4. Conclusions

Large-sized specimen wheel tracking testing on AC-13 basalt asphalt mixtures was carried out to study the development of the mixture under heavy load condition. The changing rules of deformation and skid resistance were consistent with the results of standard tests [1,6]. The following results were obtained:(1)In the service of nine years, the permanent deformation of the asphalt mixture increased slowly under low load, while developed by obvious stages under high load: increased rapidly at the initial stage, and then slowed down. The deformation growth speed of the asphalt mixture under high load would decrease due to the formation of a new stable structure after 60,000 cycles. There was a critical load between the two development models, which was related to the performance of the aggregates.(2)The asphalt mixture on the surface had been compacted, exposing the corners and edges of the aggregates after grinding approximately 5000 times. The skid resistance began to decline rapidly after this point. BPN and TD had a larger degradation range and smaller final value under higher load.(3)In terms of permanent deformation, the asphalt mixture remained effective for a long time under low load, while it failed after grinding 45,000 times under high load. The failure cumulative cycles of skid resistance under various loads was predicted by the fitting curve. The failure cumulative cycles on BPN under 0.7 MPa, 0.9 MPa, 1.1 MPa, and 1.4 MPa was 4.09 million, 1.26 million, 0.55 million, and 0.34 million, respectively. The failure cumulative cycles on TD under 0.7 MPa, 0.9 MPa, 1.1 MPa, and 1.4 MPa was 0.88 million, 0.39 million, 0.23 million, and 0.12 million, respectively. The failure on TD was faster than BPN.(4)According to the sieving result of the ground samples under heavy load, it was found that the content of 0–3 mm was almost constant, 10–16 mm decreased, while 5–10 mm and 3–5 mm rose. It could be viewed as some of the 10–16 mm particles were crushed into 5–10 mm and 3–5 mm.(5)Aggregates were crushed into smaller pieces and reformed a stable structure after high load grinding. The new aggregate gradation of 10–16 mm, 5–10 mm, 3–5 mm, and 0–3 mm was approximately 10%, 37%, 23%, and 30%, respectively.(6)According to the correlation analysis, the content of 10–16 mm had a significant correlation with the rutting performance of the mixture. There was no significant relationship between the aggregate gradation and skid resistance of the asphalt mixture. With the increase in the load, the correlation between permanent deformation and the aggregates gradually decreased.

## Figures and Tables

**Figure 1 materials-12-03814-f001:**
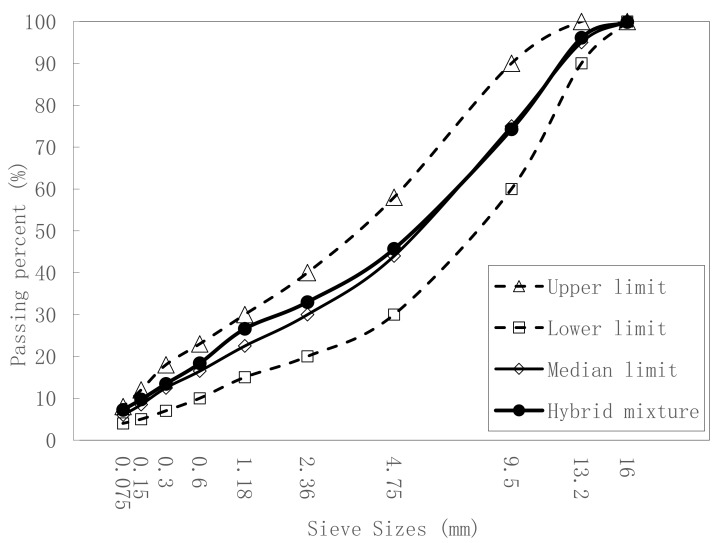
AC-13 gradation curve.

**Figure 2 materials-12-03814-f002:**
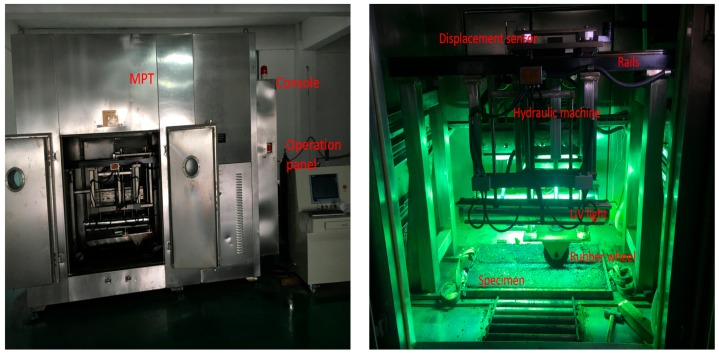
Multifunctional pavement material tester and its internal structure.

**Figure 3 materials-12-03814-f003:**
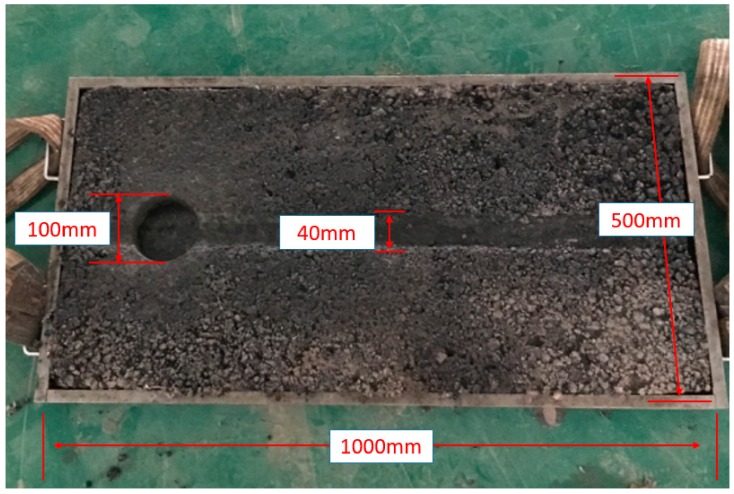
Specimen size and the core drilling.

**Figure 4 materials-12-03814-f004:**
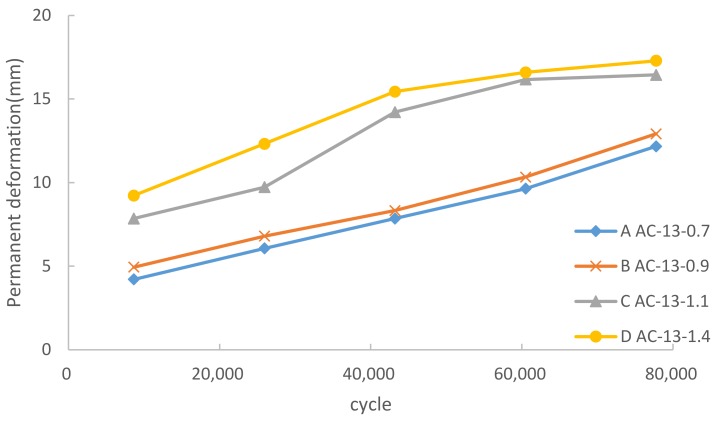
Heavy load effect on permanent deformation.

**Figure 5 materials-12-03814-f005:**
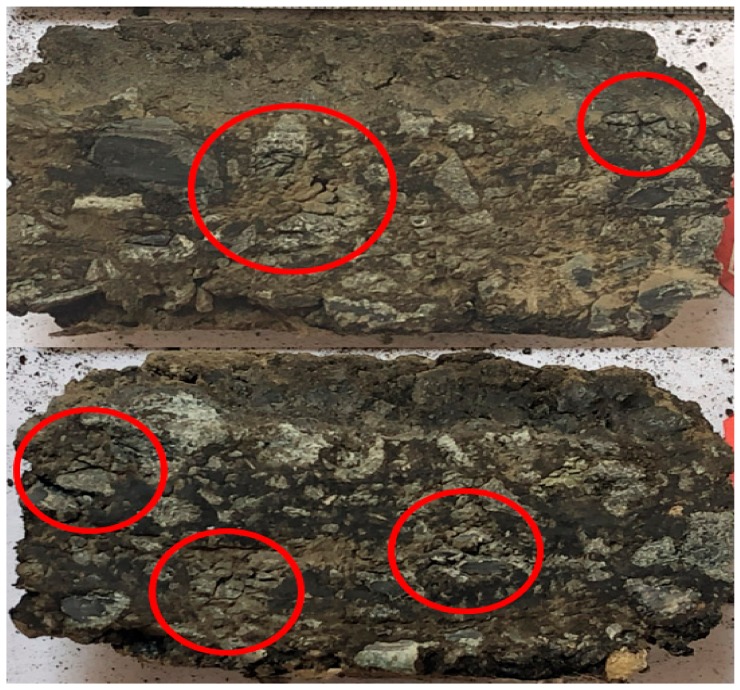
Broken part in the cross-section view of the samples.

**Figure 6 materials-12-03814-f006:**
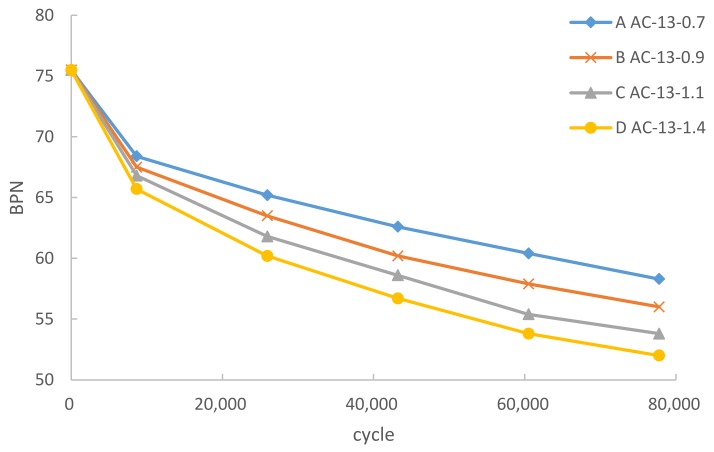
Heavy load effect on the British pendulum number.

**Figure 7 materials-12-03814-f007:**
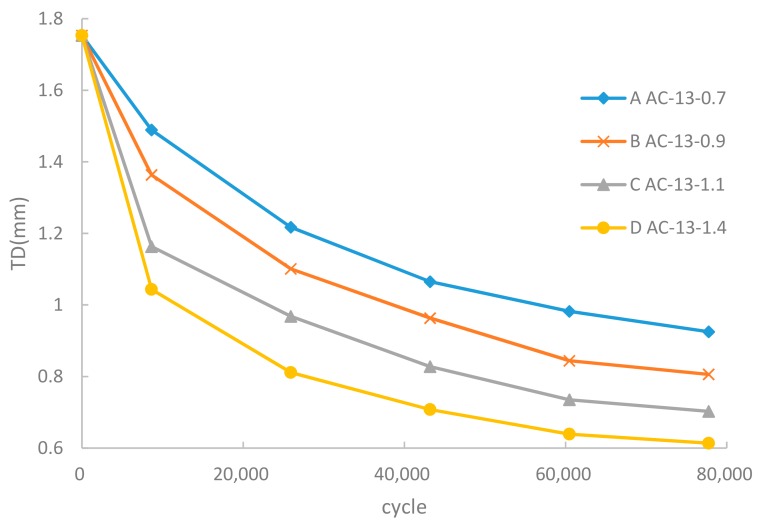
Heavy load effect on TD.

**Figure 8 materials-12-03814-f008:**
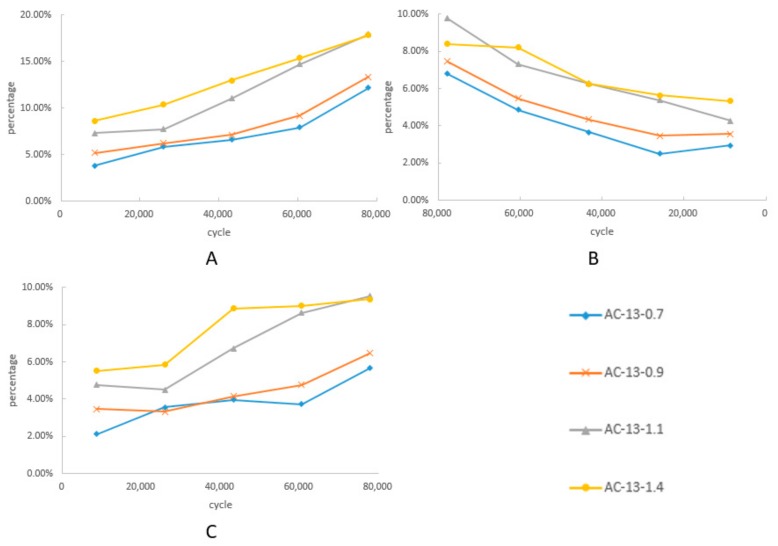
Gradation change in the ground mixture. (**A**) The loss in 10–16 mm, (**B**) the rise in 5–10 mm, (**C**) the rise in 3–5 mm.

**Figure 9 materials-12-03814-f009:**
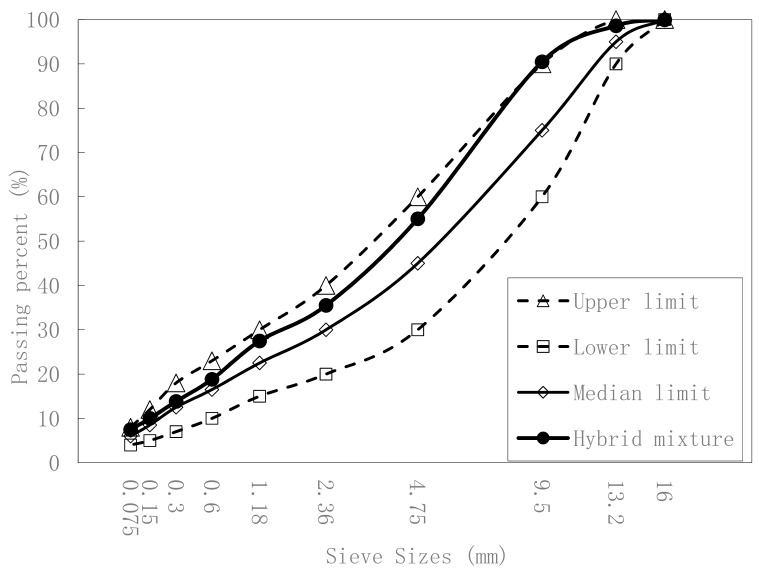
Gradation curve after heavy load.

**Table 1 materials-12-03814-t001:** Physical properties of the SBS asphalt binder.

Test Items		Results	Requirements
Penetration 25 °C, 100 g, 5s (0.1 mm)		76.2	60~80
Softening point (°C)		63.6	≥60
Ductility 5 cm/min, 5 °C (cm)		51.2	≥30
Density 15 °C (g/cm^3^)		1.031	/
Solubility (TCE) (%)		99.8	≥99
Aging 163 °C, 5 h	Mass change (%)	0.4	±1.0
	Penetration ratio 25 °C (%)	83.3	≥60
	Residual ductility 5 °C (cm)	32.3	≥20

**Table 2 materials-12-03814-t002:** Heavy load testing program for the AC-13 basalt asphalt mixture.

Series	Load/MPa	Specimen Label	Duration/h	Cumulative Cycles
A	0.7	AC-13-0.7-1	12	8640
		AC-13-0.7-2	36	25,920
		AC-13-0.7-3	60	43,200
		AC-13-0.7-4	84	60,480
		AC-13-0.7-5	108	77,760
B	0.9	AC-13-0.9-1	12	8640
		AC-13-0.9-2	36	25,920
		AC-13-0.9-3	60	43,200
		AC-13-0.9-4	84	60,480
		AC-13-0.9-5	108	77,760
C	1.1	AC-13-1.1-1	12	8640
		AC-13-1.1-2	36	25,920
		AC-13-1.1-3	60	43,200
		AC-13-1.1-4	84	60,480
		AC-13-1.1-5	108	77,760
D	1.4	AC-13-1.4-1	12	8640
		AC-13-1.4-2	36	25,920
		AC-13-1.4-3	60	43,200
		AC-13-1.4-4	84	60,480
		AC-13-1.4-5	108	77,760

**Table 3 materials-12-03814-t003:** Sieving result of ground asphalt mixture.

Specimen Label	10–16 mm	5–10 mm	3–5 mm	0–3 mm and Filler
AC-13	28%	28%	14%	30%
AC-13-0.7-1	24.18%	30.95%	16.11%	28.75%
AC-13-0.7-2	22.16%	30.51%	17.57%	29.76%
AC-13-0.7-3	21.42%	31.65%	17.95%	28.98%
AC-13-0.7-4	20.09%	32.86%	17.71%	29.34%
AC-13-0.7-5	15.82%	34.80%	19.67%	29.71%
AC-13-0.9-1	22.80%	31.57%	17.46%	28.17%
AC-13-0.9-2	21.77%	31.46%	17.32%	29.45%
AC-13-0.9-3	20.86%	32.34%	18.16%	28.64%
AC-13-0.9-4	18.82%	33.48%	18.76%	28.94%
AC-13-0.9-5	14.66%	35.46%	20.45%	29.43%
AC-13-1.1-1	20.68%	32.26%	18.75%	28.31%
AC-13-1.1-2	20.28%	33.38%	18.51%	27.83%
AC-13-1.1-3	16.97%	34.27%	20.73%	28.03%
AC-13-1.1-4	13.29%	35.30%	22.63%	28.78%
AC-13-1.1-5	10.05%	37.76%	23.53%	28.65%
AC-13-1.4-1	19.38%	33.34%	19.51%	27.77%
AC-13-1.4-2	17.62%	33.65%	19.85%	28.89%
AC-13-1.4-3	15.01%	34.26%	22.87%	27.86%
AC-13-1.4-4	12.61%	36.22%	23.02%	28.16%
AC-13-1.4-5	10.17%	36.40%	23.37%	30.06%

**Table 4 materials-12-03814-t004:** Correlation between gradation and mixture performance.

Load/MPa	Aggregate/mm	Permanent Deformation		BPN		TD	
		PCC	Sig.	PCC	Sig.	PCC	Sig.
0.70	10–16	0.972	0.006	−0.934	0.200	−0.857	0.063
	5–10	0.938	0.018	−0.883	0.047	−0.762	0.134
	3–5	0.923	0.025	−0.906	0.034	−0.877	0.051
0.90	10–16	0.972	0.006	−0.889	0.044	−0.826	0.085
	5–10	0.957	0.011	−0.868	0.056	−0.794	0.109
	3–5	0.950	0.013	−0.861	0.061	−0.783	0.117
1.10	10–16	0.926	0.024	−0.929	0.023	−0.896	0.040
	5–10	0.882	0.048	−0.928	0.023	−0.892	0.042
	3–5	0.948	0.014	−0.922	0.026	−0.897	0.039
1.40	10–16	0.950	0.013	−0.969	0.007	−0.916	0.029
	5–10	0.882	0.048	−0.912	0.031	−0.845	0.072
	3–5	0.957	0.011	−0.927	0.023	−0.901	0.037

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
