# Peer review of "Performance Degradation of Large-Sized Asphalt Mixture Specimen under Heavy Load and Its Affecting Factors Using Multifunctional Pavement Material Tester"

_materials, 2019, doi:10.3390/ma12233814_

Round 1
Reviewer 1 Report
The authors investigated the changes in the performance of pavement under the heavy load.
Proofreading and improving the language of the paper is necessary. There are several grammatical and format errors. Please proofread the article carefully and also use my comments as a guidance.
Comments:
Line 28: Please insert space before citations and combine citation numbers. [1-2]
Line 32: Use space between unit and number
Line 41: TD is related to macro texture and BPN is related to micro texture. Is the order mentioned in the text correct?
Line 52: aggregates instead of “aggregate”
Line 53: Shouldn’t the numbers be in percentage?
Figure 1: Remove borders of the figures and use the same format for all the graphs. Match Figures 1 and 4 formats. (Colors, line widths and legends, etc.)
Table 1: The "Aging" row of the table is mixed.
Line 63: Remove dot before (Figure 2)
Line 72 and 72: Rewrite the sentence.
Figure 3: It is not clear that 40 mm is the thickness of asphalt sample. Please put the dimension sign somewhere else.
Line 98: WLF : Explain all acronyms first time they appear in your article.
Line 98 and 99: Rewrite the sentence.
Equation 1: The font of formula is large compared to text.
Line 101: Match font of text and formula.
Line 104: "It could be figured out 103 that 1 day in the MPT equaled 4.65 day in real condition. " Please mention what the assumed values are to measure it and reference appropriately to any reference used in the calculations.
Figure 4: Remove Border of the image and match its format with other images. Put y axis unit in parentheses.
Line 120: At least explain once in the text that what A, B, C and D are.
Lines 139-143: Rewrite sentences and check for spell.
Figure 5: Do not use title inside the graph. Please explain them in the caption.
Figure 6: Correct y-axis label.
Lines 190 and 191:
“As the grinding proceed, the 10-16mm part was on a decline curve 190 and the 5-10mm and 3-5 parts were on the rise.”
This conclusion is not clear. Please rewrite the sentence.
Figure 7: Please remove figure title and explain it in the caption.
y axis label: percentage of what?
Also, please rearrange the graphs A, B and C and the legend position.
Line 241: “finial” : Typo
Reviewer 2 Report
Review of the paper "Performance degradation of large-sized asphalt mixture specimen under heavy load and its affecting factors using multifunctional pavement material tester”
In this paper, the authors presented the effect of overloading on the change of asphalt mix properties.
In my point of view, the authors investigate an interesting subject with good impacts to the field.
However, the paper needs moderate/major revision before it is in the good shape for publication in MDPI "Materials".
Therefore, in the revised version I would like to see the following comments/concerns addressed and then I will provide my final recommendation of this work:
In line 31-35: Please complete with the following data: the replacement diameter of the car wheel contact with the surface.
In point 2. It would be helpful to present the percentage of asphalt mixtures components.
in section 2. Materials and experimental program, it should be described why the authors chose basalt aggregate. In addition, information on aggregate in the PSV (polished stone value) parameter should be completed. Basalt aggregate has low PSV polishing resistance. The authors should consider whether, for comparison, they should not perform the same tests for another aggregate with a larger PSV, e.g. granite.
in point 2. Materials and experimental program - what layer is AC-13 planned for? I understand that for the surface layer but it is good to indicate this in the article. in point 3.1. Permanent deformation results - (line 125 - 128) The authors write about the deformation (mm) was shelled in the initial phase of the test, while the test result presented in Figure 4 presents a constant and gradual increase in the entire test range (up to 80,000 cycles) - please comment.
The increase in defrmation can be described using WTS (wheel-tracking rate)
WTS = (h80000-h40000) / 4
where h - is the depth of deformation in the cycle.
line 184 - 186 "The sieving results of ground mixture are shown in the Table 3. Due to the loss of fine aggregate and mineral powder filler during dissolution and filtration, the percentage of 0-3mm part was scaled up to compensate." - please explain why loss of fine aggregate and mineral powder filler. Please provide a comparison of the grain size curve before the test (Fig. 1) to that obtained after the test. Is still between the upper and lower limits.
Reviewer 3 Report
Interesting paper
Please, provide a flowchart of the selected approach. More details are required in the experimental results section. Why combining all these tests together? It results difficult to distinguish the different impacts of every single test. How was the air voids content controlled? Section 3.4 need to be expanded and detailed. Conclusions can be improved. Please, double-check English for typos and spelling. Some paragraphs are hard to read. The paper is valuable from a practical point of view to engineers and practitioners.Author Response
Please see the attachment

Reviewer 4 Report
Dear authors
thank you for your article regarding the relationship between the number of rutting cycles, texture, skid resistance and composition when subjected to a heavy load.
Unfortunately, I am not convinced about the results shown in this paper. I would like to see some of the results that are claimed in the paper, such as the removal of the bitumen film after a certain number of cycles. The proposed method is denominated as "heavy loaded", but a comparison of the suggested method compared to other standardized methods and actual heavy axle loads is needed. Furthermore, a statistical correlation is established between the aggregate fractions and mixture performance. It is stated there as well, that the larger fraction is "broken" into two smaller fractions as a result of the heavy load. This seems unrealistic as the used mixture consists of a "sand skeleton" (not a stone skeleton as in an SMA), so it seems unlikely that the stones are into direct contact and will break. It is necessary to make sure that the used mixing and compaction methods do not lead to the inhomogeneity of the mixture. Figure 3 does look inhomogeneous to me ...
Furthermore, it is unclear why you decided to use plates of 1x0,5 m as you are only interested in the middle part where you drill cores. Why not use the standard sample sizes, where mixing and compaction might be better? Therefore I see no reason for the "large-sized asphalt specimen" as mentioned in the title. In general, the way you have extracted the cores from these plates in unclear (how long does it take to extract the cores and how does it affect the temperature of the specimen, what influence does the filling of the extracted core have, which kind of mixture did you use for this and here they compacted and to which height, ...)
Correlations between texture depth and skid resistance have been investigated before as apparent from your own literature study, so why are they not used to check whether your own findings are similar/ as can be expected? Another aspect is one of your conclusions: The asphalt film on the surface of asphalt mixture had been worn away, exposing the surface 239 of the aggregate after approximately 5000 times of grinding. The skid resistance began to decline 240 rapidly after this point. In practice, the removal of this bitumen film on the road is needed, so in situ skid resistance measurements are only performed after 4-6 weeks. It is surprising that you find that the skid resistance decreases after this removal ...
Regarding the literature study, I see no papers from Materials or Applied Sciences, so why are you submitting your paper here? It also leads to some publication bias as you seem to have focussed mostly on Construction and Building Materials and RMPD. Furthermore, in the Results section, there are almost no references, so no discussion on how your results correspond to the results of previous works. This is the most important aspect of the literature study ...
Overall, I question the validity and scientific soudness of the results and the main conclusions. Therefore, my advise is to reject this paper until more scientific proof is given.
My other detailed comments/remarks/suggestions can be found in the attached pdf-file.

Round 2
Reviewer 3 Report
The authors addressed the commentsAuthor Response
The authors addressed the comments
Reviewer 4 Report
Dear authors
I summarized my main objections in this web form, but the attached pdf document contained a lot more comments/suggestions/improvements (again attached to this review).
I really get the idea that you did not take a look at this document as a lot of minor comments could have been dealt with easily.
You mention in your rebuttal that you used over 40 samples, so I am really surprised to see not even a mention of a standard deviation or other statistical analyses.
Overall the changes that were done in the paper are very minor and are not satisfactory regarding my main concerns.
with kind regards

Round 3
Reviewer 4 Report
Dear authors
Significant improvement of the paper, where more and detailed proof (e.g. new Figure 5) was given, some errors were corrected (e.g. axle load instead of vehicle load) and more explanation was given about the methodology.
Some final comments are given below, that can be dealt with by the authors without further need for a review:
Language and editing errors should be checked the last time, but I leave that responsibility to MDPI References: some things went wrong in this version where some references are missing brackets, e.g. 6 instead of [6] Below equation (1): line 113: what is the value 25 doing there (or a reference?); line 114: do you mean the average maximum surface temperature?; line 119: traffic volume more than 1500 times (more than what? Unclear) Line 189: to my knowledge SPSS = Statistical Package for the Social Sciences, so your abbreviation surprises me Lines 218-221: wrong reference to Figure 7 instead of Figure 8 + include information about the subfigures in the caption of Figure 8 Lines 265-267: I think you will also have a very nice correlation between TD and BPN? Maybe something to mention shortly (no new table)? Reply to my comments: Axle load spectrum does interest me, but maybe not other readers. If you include it, it would be nice to know how it was determined as well (how many roads and for how long, or based on measurements of single trucks?) ==> where can I find more info on this aspect? I am working on heavy loading and axle load spectra as well. Figures shown the extracted core and the one with the corners cut off, can also help to understand how you tested everything. Consider to include them in the final versionkind regards
Author Response
Dear reviewer,
Thanks for your suggestions. Some minor revisons have been added to the article according to your advisings. Here are the answers to your questions.
1. line 114: do you mean the average maximum surface temperature
This temperature is the average surface temperature weighted by traffic volume. that means the rutting created by vehicles all year round is the same as the rutting created by vehicles at this temperature.
2. Line 189: the full name of SPSS
SPSS company changed its full name from " Statistical Package for the Social Sciences" to "Statistical Product and Service Solutions" in 2000. (you can refer to Wiki)
3. where can I find more info on this aspect?
Some of the data in my paper are from traffic surveys across China and some papers published in China. The data about heavy loading are mainly from the traffic surveys of Guangdong province, China. ( that is a prosperus province in south China with developed road transportation) Maybe you can follow the annual reports issued by China's Ministry of Transport and provincial transport departments. I don't know if those papers have been published in English. Overall, I think China's traffic data is very useful for studying heavy traffic, because the problem of overloading is very serious in China.
kind regards